# PLANNING WITH UNCERTAINTY: DEEP EXPLORATION IN MODEL-BASED REINFORCEMENT LEARNING

## ABSTRACT

Deep model-based Reinforcement Learning (RL) has shown super-human performance in many challenging domains. Low sample efficiency and limited exploration remain as leading obstacles in the field, however. In this paper, we demonstrate deep exploration in model-based RL by incorporating epistemic uncertainty into planning trees, circumventing the standard approach of propagating uncertainty through value learning. We evaluate this approach with the state of the art model-based RL algorithm MuZero, and extend its training process to stabilize learning from explicitly-exploratory trajectories. In our experiments planning with uncertainty is able to demonstrate deep exploration with standard uncertainty estimation mechanisms, and with it provide significant gains in sample efficiency in hard-exploration problems.

## 1 INTRODUCTION

In February 2022, state of the art performance in video compression on YouTube videos has been achieved with the algorithm MuZero (Mandhane et al., 2022; Schrittwieser et al., 2020), setting a new milestone in successful deployments of reinforcement learning (RL). MuZero is a deep model-based RL algorithm that learns an abstracted model of the environment through interactions, and uses it for planning. While able to achieve state of the art performance in extremely challenging domains, MuZero is limited by on-policy exploration that relies on random action selection. Effective, informed exploration is crucial in many problem settings (O'Donoghue et al., 2018), and can induce up to exponential gain in sample efficiency (Osband et al., 2016).

Standard approaches for exploration rely on estimates of *epistemic uncertainty* (uncertainty that is caused by a lack of information, Hüllermeier & Waegeman, 2021) to drive exploration into under-unexplored areas of the state-action space (Bellemare et al., 2016; Sekar et al., 2020) or the learner's parameter space (Russo et al., 2018). These approaches often incorporate the uncertainty into value-learning as a non-stationary reward bonus (Oudeyer & Kaplan, 2009) or to directly approximate the total uncertainty in a value-like prediction propagated over future actions (O'Donoghue et al., 2018) to achieve exploration that is *deep*. We refer to deep exploration as exploration that is 1) directed over multiple consecutive time steps, as well as 2) farsighted with respect to information about future rewards, following the definition by (Osband et al., 2016). Having access to deep exploration enables the agent to aim for (through farsightedness) and reach (through directness over multiple time steps) areas that are both attractive to explore as well as far (in number of transitions) from its initial or current state.

While propagating the uncertainty from future actions through the value function enables propagation over a far horizon, it also introduces several problems. First, the values become non-stationary with the uncertainty bonus, introducing potential instability into the value learning. Second, the horizon of the propagation is limited in the number of training steps: uncertainty in states that are far from the initial state will only propagate to the initial state after sufficiently many training steps, and not immediately. Third, the propagation speed is directly correlated with number of training steps, and as a result uncertainty from encountered high-uncertainty areas of the state space that are not trained often will not (or barely) propagate. To overcome these problems, this paper proposes to propagate uncertainty through the planning-tree of model-based RL instead of a neural-network value function. This facilitates the decoupling of the propagation from the learning process by

propagating the uncertainty during online inference, and addresses the challenges originating from propagating the uncertainty through value learning.

To demonstrate that propagating uncertainty in the online planning-trees of model-based methods like MuZero can provide deep exploration, this paper's contribution is divided into three parts. First, we propose a framework for propagating epistemic uncertainty about the planning process itself through a planning tree, for example, in Monte Carlo Tree Search (MCTS, see Browne et al., 2012, for an overview) with learned models. Second, we propose to harness planning with uncertainty to achieve deep exploration in the standard RL setting (in difference to prior approaches, Sekar et al., 2020, that required a pre-training phase to explore), by modifying the objective of an online planning phase with epistemic uncertainty, which we dub online planning to explore, or OP2E. Third, to stabilize learning from off-policy exploratory decisions in MuZero, we extend MuZero's training process by splitting training into exploration episodes and exploitation episodes, and generating the value and policy targets differently depending on the type of episode. We conduct experiments against hard-exploration tasks to evaluate the capacity of OP2E to achieve deep exploration. In addition, we conduct an ablation study to evaluate the individual effects of the different extensions we propose to training from explicitly exploratory trajectories. In our experiments OP2E was able to significantly outperform vanilla MuZero in hard-exploration tasks, demonstrating deep exploration resulting in significant gains in sample efficiency. Our ablation study points out the potential value of discerning between positive and negative exploration trajectories and generating policy targets accordingly, as well as the resilience of n-step value targets to the presence of strongly off-policy trajectories.

This paper is organized as follows: Section 2 provides relevant background for model-based RL, MuZero and epistemic uncertainty estimation in deep learning. Section 3 describes our contributions, starting with the framework for uncertainty propagation, followed by our approach for modifying planning with uncertainty to achieve deep exploration (OP2E) and finally the extensions proposed to stabilize learning from exploratory decisions. Section 4 evaluates OP2E against two hard-exploration tasks in comparison with vanilla MuZero and presents and ablation study of the different extensions to learning from exploratory decisions. Section 5 discusses related work, and Section 6 concludes the paper and discusses future work.

## 2 BACKGROUND

### 2.1 REINFORCEMENT LEARNING

In reinforcement learning (RL), an agent learns a behavior policy $\pi(a|s)$ through interactions with an environment, by observing states (or observations), executing actions and receiving rewards. The environment is represented with a Markov Decision Process (MDP, Bellman, 1957), or a partially-observable MDP (POMDP, Åström, 1965). An MDP $M$ is a tuple: $\mathcal{M} = \langle S, A, \rho, P, R \rangle$, where $S$ is a set of states, $A$ a set of actions, $\rho$ a probability distribution over the state space specifying the probability of starting at each state $s \in S$, $R : S \times A \to \mathbb{R}$ a bounded reward function, and $P : S \times A \times S \to [0, 1]$ is a transition function, where $P(s'|s, a)$ specifies the probability of transitioning from state $s$ to state $s'$ after executing action $a$. In a POMDP $\mathcal{M}' = \langle S, A, \rho, P, R, \Omega, O \rangle$, the agent observes observations $o \in \Omega$. $O : S \times A \times \Omega \to [0, 1]$ specifies the probability $O(o|s, a)$ of observing a possible observation $o$.

In model based reinforcement learning (MBRL) the agent learns a model of the environment through interactions, and uses it to optimize its policy, often through *planning*. In deep MBRL (DMBRL) the agent utilizes deep neural networks (DNNs) as function approximators. Many RL approaches rely on learning a state-action *Q-value function* $Q^\pi(s, a) = R(s, a) + \gamma \mathbb{E}[V^\pi(s')|s' \sim P(\cdot|s, a)]$ or the corresponding state *value function* $V^\pi(s) = \mathbb{E}[Q^\pi(s, a)|a \sim \pi(\cdot|s)]$, which represents the expected return from starting in state $s$ (and possibly action $a$) and then following policy $\pi$.

### 2.2 MONTE CARLO TREE SEARCH

MCTS is a planning algorithm that constructs a planning tree with the current state $s_t$ at its root, to estimate the objective: $\arg\max_a Q^\pi(s_t, a)$. The algorithm repeatedly performs the following four steps: *trajectory selection*, *expansion*, *simulation* and *backup*. Starting from the root node $n_0 \equiv s_t$, the algorithm selects a trajectory in the existing tree based on the averaged returns $q(n_k, a)$

experienced in past trajectories selecting the action $a$ in the same node $n_k$, and one of many search heuristics, such as an Upper Confidence Bound for Trees (UCT, Kocsis & Szepesvári, 2006):

$$a_k = \arg\max_{a \in A} q(n_k, a) + 2C_p \sqrt{\frac{2\log(\sum_{a'} N(n_k, a'))}{N(n_k, a)}} \tag{1}$$

where $N(n_k, a)$ denotes how many times action $a$ has been executed in node $n_k$. The trajectory selection proceeds iteratively until it arrives at a leaf node, that has yet to be expanded in the tree. MCTS expands the node and estimates its initial value with a rollout from a random policy. However, Alpha Go/Zero (Silver et al., 2016; 2017; 2018) uses a value function $v(n_k)$ that is approximated by a neural network to estimate the leaf's value and chooses potential actions $a$ to expand first with another approximation of the MCTS policy $\pi(a|n_k)$. Last, MCTS propagates the return (discounted reward for visited nodes plus leaf's value) back along the planning trajectory, updating the averaged returns $q(n_k, a_k)$, all the way up to the root of the tree. By choosing different trajectory selection heuristics, MCTS can be used to optimize for different objectives. At the root of the tree, the value $V^\pi(s_t)$ of current state $s_t$ is estimated based on the averaged returns experienced through every action $a$, and averaged over the actions:

$$V^\pi(s_t) \approx \sum_{a \in A} \frac{q(n_0, a) N(n_0, a)}{\sum_{a' \in A} N(n_0, a')} =: v_t^\pi \tag{2}$$

## 2.3 MuZero

MuZero is a MBRL algorithm that learns an abstracted model of the environment from interactions, and uses it for planning with MCTS. The model learned by MuZero consists of 5 functions that are approximated with neural networks: the representation function $\hat{s}_0 = g(o_t)$, the transition function $\hat{s}_{k+1} = f(\hat{s}_k, a_k)$, the expected-reward function $\hat{r}_k = r(\hat{s}_k, a_k)$, the value function $\hat{v}_k = v(\hat{s}_k)$ and policy function $p(\hat{s}_k) = \pi(a|\hat{s}_k)$. In practice, the representation and dynamics are only used as node-representations $\hat{s}_k = n_k$ in the MCTS planning tree, and only need to represent the states of the environment as far as they are useful for the MCTS. The variation of MCTS used by MuZero approximates the value of a leaf-node in the expansion step with a direct predictions from the value function $v$ instead of Monte-Carlo simulations. MCTS is used in MuZero for two purposes: first, for inference of the best action to execute in the environment, and second to generate targets for the policy and value functions. MuZero is trained with on-policy value and policy targets. The policy targets are the relative visitations numbers to each child of the root node, and the value targets are $n$-step TD targets:

$$v_t^{target} = \sum_{i=0}^{n-1} \gamma^i r_{t+i}^\pi + \gamma^n v_{t+n}^\pi,$$

Where $\pi$ is the policy that was executed in the environment which was induced by the MCTS at every time step. With the Reanalyse variation of MuZero (Schrittwieser et al., 2021), the policy and value targets are constantly updated in the replay buffer based on new planning trees that are computed by Reanalyse, turning MuZero Reanalyse into a (mostly) off-policy algorithm. The updated value targets are still $n$-step targets that use the sum of rewards $\sum_{i=0}^{n-1} \gamma^i r_{t+i}^\pi$ observed in the environment following policy $\pi$. This prevents MuZero Reanalyse from being a completely off-policy algorithm and makes it challenging to learn from explicitly-exploratory trajectories that do not follow the current exploitation policy.

## 2.4 Epistemic Uncertainty Estimation in Deep Learning

Defining, quantifying and estimating predictive epistemic uncertainty is an active field of machine learning research that encompasses many approaches and many methods (Hüllermeier & Waegeman, 2021; Jain et al., 2021). In this work we assume epistemic uncertainty can be represented with probability theory, and estimated as the variance in a probability distribution of predictions that are consistent with the observations $\text{Var}(X|o_t) = \mathbb{V}_X[X|o_t]$, which is a standard approach for epistemic uncertainty quantification. This allows us to investigate the propagation of uncertainty in predictions of a node $n_k$ in a planning tree with respect to observations and a trajectory of actions through the lens of variance propagation $\mathbb{V}_{X_k}[X_k|o_t, a_{0:k}]$ which is well studied.

To directly estimate the epistemic variance in the local predictions of value and reward used by MuZero in planning we use two methods: deep ensembles with prior functions (Osband et al.,

2018) and state-visitation counting. Ensembles are scalable to arbitrarily large state spaces as well as computationally inexpensive for small ensemble sizes. The epistemic uncertainty estimation ensembles provide however is not necessarily reliable. As an additional method that is able to provide reliable estimates of epistemic uncertainty we use state visitation counting. This method estimates epistemic uncertainty directly based on the number of times each state or state-action pair has been visited. While a toy method, visitations counting enables us to evaluate the approach in the presence of both reliable and unreliable uncertainty estimators. Additional details regarding applying these methods to the functions used by MuZero can be found in appendix B.

## 3 Contributions

Our approach modifies MCTS to plan for an exploratory objective rather than an exploitatory objective, i.e., to predict the best action that can be taken in the environment for gathering relevant information rather than for highest expected return. To apply this approach with MuZero as a use-case, we take three steps. First, we propose approximations for uncertainty propagated through a planning tree, forward and back (Section 3.1). Second, we propose to modify the UCT operator with the propagated uncertainty to optimize action selection for exploration with a so-called optimistic objective (Section 3.2). Third, we propose to extend the training process of MuZero by introducing a distinction between exploration and exploitation episodes and by modifying the target generation from exploratory trajectories. The target generation is extended to discern between exploration trajectories that resulted in returns that are higher than the agent's value approximations for the same initial state, and trajectories that did not. For exploration trajectories that resulted in improved returns, the exploratory decisions become the targets from the learning trajectory. Otherwise, the agent's exploitatory approximations provide the target (Section 3.3).

### 3.1 Propagating Uncertainty in a Planning Tree

Selecting a path $n_{0:T}$ through a decision tree is equivalent to choosing a sequence of $T$ actions $a_{0:T-1}$ that end up in a leaf node $n_T$. In standard MCTS a deterministic model predicts the encountered rewards $r_k$ in nodes $n_k, 0 \leq k < T$, and the estimated value $v_T$ at the leaf $n_T$. These are used to update the $n$-step discounted return $\nu_k$ of each node $n_k$ on the selected path:

$$\nu_k \quad := \quad \sum_{i=k}^{T-1} \gamma^{i-k} r_i + \gamma^T v_T \quad = \quad r_k + \gamma \nu_{k+1}, \qquad 0 \leq k < T, \qquad \nu_T = v_T \qquad (3)$$

If the model that produced states, rewards and values cannot be fully trusted, $r_k$ and $v_T$ can be formulated as random variables in a Markov chain that is connected by random state-variables $s_k, 0 \leq k \leq T$. These correspond to the nodes on the path, the corresponding states, or their representations, but not necessarily to the output $\hat{s}_k$ of any MuZero transition model (Section 2.3).

In exploration, paths that lead to under-explored states, or states where the model is not reliable yet, should be incentivized. In line with optimistic exploration literature, we incentivize choosing actions in the environment associated with paths in the planning tree that have *uncertain* returns $\nu_0$ in order to both improve the model as well as find high-reward interactions. For this we need to estimate the variance $\mathbb{V}_\nu[\nu_0|o_t, a_{0:T-1}]$ of the return along a selected path $a_{0:T-1}$, starting with observation $o_t$. To improve readability, we will omit in the following the condition on actions and observation, for example, $\mathbb{V}_\nu(\nu_0) \equiv \mathbb{V}_\nu[\nu_0|o_t, a_{0:T-1}]$.

We will begin by deriving the mean and variance of the distribution of state-variables in the Markov chain for a given sequence of actions $a_{0:T-1}$. Let us assume we are given a differentiable transition function $f(s_k, a_k) := \mathbb{E}_{s_{k+1}}[s_{k+1}|s_k, a_k] \in \mathbb{R}^{|S|}$, which predicts the average next state, and a differentiable uncertainty function $\Sigma(s_k, a_k) := \mathbb{V}_{s_{k+1}}[s_{k+1}|s_k, a_k] \in \mathbb{R}^{|S| \times |S|}$ that yields the covariance matrix of the distribution. We will use an epistemic uncertainty measure to compute the latter, but in principle these functions could represent any distribution, containing any kind of uncertainty, epistemic or otherwise. We assume that the mean $\bar{s}_0$ of the first state-variable $s_0$ is given as an encoding function $\bar{s}_0 = \mathbb{E}_{s_0}[s_0|o_t] = g(o_t)$, like in MuZero. The mean $\bar{s}_{k+1}$ of a later state-variable $s_{k+1}$ can be approximated with a 1st order Taylor expansion around the previous mean

$\bar{s}_k := \mathbb{E}_{s_k}[s_k]$:

$$\begin{aligned}
\bar{s}_{k+1} &:= \mathbb{E}_{s_{k+1}}[s_{k+1}] = \mathbb{E}_{s_k}[\mathbb{E}_{s_{k+1}}[s_{k+1}|s_k, a_k]] = \mathbb{E}_{s_k}[f(s_k, a_k)] \quad (4) \\
&\approx \mathbb{E}_{s_k}[f(\bar{s}_k, a_k) + (s_k - \bar{s}_k)^\top \nabla_s f(s, a_k)|_{s=\bar{s}_k}] = f(\bar{s}_k, a_k).
\end{aligned}$$

To approximate the covariance $\bar{\Sigma}_{k+1} := \mathbb{V}_{s_{k+1}}[s_{k+1}]$, we need the *law of total variance* where for two random variables $x$ and $y$ holds $\mathbb{V}_y[y] = \mathbb{E}_x[\mathbb{V}_y[y|x]] + \mathbb{V}_x[\mathbb{E}_y[y|x]]$ (see appendix A for a proof in our notation), and again a 1st order Taylor approximation around the previous mean state $\bar{s}_k$:

$$\begin{aligned}
\bar{\Sigma}_{k+1} &:= \mathbb{V}_{s_{k+1}}[s_{k+1}] = \underbrace{\mathbb{E}_{s_k}[\mathbb{V}_{s_{k+1}}[s_{k+1}|s_k]]} + \underbrace{\mathbb{V}_{s_k}[\mathbb{E}_{s_{k+1}}[s_{k+1}|s_k]]} \quad (5) \\
&\approx \Sigma(\bar{s}_k, a_k) + J_f(\bar{s}_k, a_k)\,\bar{\Sigma}_k\,J_f(\bar{s}_k, a_k)^\top.
\end{aligned}$$

Note that here $f(s_k, a_k) - \mathbb{E}_{s_k}[f(s_k, a_k)] \approx (s_k - \bar{s}_k)^\top \nabla_s f(s, a_k)|_{s=\bar{s}_k} =: (s_k - \bar{s}_k)^\top J_f(\bar{s}_k, a_k)^\top$, where $J_f(\bar{s}_k, a_k)$ denotes the Jacobian matrix of function $f$ at state $\bar{s}_k$ and action $a_k$. Using these state statistics, we can derive the means and variances of causally connected variables like rewards and values. We assume that the conditional reward distribution has mean $r(s_k, a_k) := \mathbb{E}_{r_k}[r_k|s_k, a_k]$ and variance $\sigma_r^2(s_k, a_k) := \mathbb{V}_{r_k}[r_k|s_k, a_k]$, and that the conditional value distribution has mean $v(s_T) := \mathbb{E}_{v_T}[v_T|s_T]$ and variance $\sigma_v^2(s_T) := \mathbb{V}_{v_T}[v_T|s_T]$. Analogous to above we can derive:

$$\bar{r}_k := \mathbb{E}_{r_k}[r_k] \approx r(\bar{s}_k, a_k), \qquad \mathbb{V}_{r_k}[r_k] \approx \sigma_r^2(\bar{s}_k, a_k) + J_r(\bar{s}_k, a_k)\,\bar{\Sigma}_k\,J_r(\bar{s}_k, a_k)^\top, \quad (6)$$

$$\bar{v}_T := \mathbb{E}_{v_T}[v_T] \approx v(\bar{s}_T), \qquad \mathbb{V}_{v_T}[v_T] \approx \sigma_v^2(\bar{s}_T) + J_v(\bar{s}_T)\,\bar{\Sigma}_T\,J_v(\bar{s}_T)^\top. \quad (7)$$

If we assume that $r_k$ and the $n$-step return $\nu_{k+1}$ from Equation 3 are independent, we can compute

$$\mathbb{E}_{\nu_k}[\nu_k] = \mathbb{E}_{r_k \nu_{k+1}}[r_k + \gamma \nu_{k+1}] = \mathbb{E}_{r_k}[r_k] + \gamma \mathbb{E}_{\nu_{k+1}}[\nu_{k+1}], \quad \mathbb{E}_{\nu_T}[\nu_T] = \mathbb{E}_{v_T}[v_T], \quad (8)$$

$$\mathbb{V}_{\nu_k}[\nu_k] = \mathbb{V}_{r_k \nu_{k+1}}[r_k + \gamma \nu_{k+1}] = \mathbb{V}_{r_k}[r_k] + \gamma^2 \mathbb{V}_{\nu_{k+1}}[\nu_{k+1}], \quad \mathbb{V}_{\nu_T}[\nu_T] = \mathbb{V}_{v_T}[v_T]. \quad (9)$$

We can therefore compute the variance $\mathbb{V}_\nu[\nu_0|o_t, a_{0:T-1}]$ using one forward pass through the selected path computing all $\bar{s}_k$ and $\bar{\Sigma}_k$, starting with the observation encoding $\bar{s}_0 = g(o_t)$, and one backwards pass computing $\mathbb{E}_{\nu_k}[\nu_k]$ and $\mathbb{V}_{\nu_k}[\nu_k]$.

As stated, the above derivation applies to any type of uncertainty that is encapsulated in the distribution that is propagated. For the purpose of driving exploration using uncertainty from sources that are epistemic, we employ epistemic uncertainty estimators such as deep ensembles to estimate the (co-)variances $\Sigma(s_k, a_k), \sigma_r^2(s_k, a_k), \sigma_v^2(s_T)$. To avoid modifying MuZero's planning further and allow for this approach to remain modular from the specific algorithm it is incorporated into, we interpret MuZero's representation $g$, dynamics $f$, value $v$ and reward $r$ functions as outputting the conditional means $\bar{s}_0 := \hat{s}_0, \bar{s}_k := \hat{s}_k, \bar{v}_T := \hat{v}_T, \bar{r}_k := \hat{r}_k$, respectively.

## 3.2 Planning for Exploration

The UCT operator of MCTS takes into account uncertainty *sourced in the planning tree* in the form of a node visitation count (equation 1), to drive exploration *inside* the planning tree. To drive exploration in the environment, we add the environmental epistemic uncertainty into the UCT in a similar manner, as the averaged standard deviation:

$$a_{k-1} := \arg\max_a q(n_{k-1}, a) + 2C_p \sqrt{\frac{2\log(\sum_{a'} N(n_{k-1}, a'))}{N(n_{k-1}, a)}} + C_\sigma \sqrt{\frac{\sigma_\nu^2}{N(n_{k-1}, a)}}. \quad (10)$$

$C_\sigma \geq 0$ is a constant that can be tuned per task to encourage more or less exploration. The term $\sigma_\nu^2 = \sum_{i=1}^{N(n_{k-1}, a)} \mathbb{V}_{\nu_k^i}[\nu_k^i]$ sums the variances computed individually at every backup step $i$ through node $n_k$. At each backup step $i$, with actions $a_k^i$, state means $\bar{s}_k^i$ and covariances $\bar{\Sigma}_k^i$, the variance $\mathbb{V}_{\nu_k^i}[\nu_k^i]$ is approximated based on equations 9 and 6:

$$\mathbb{V}_{\nu_k^i}[\nu_k^i] \approx \sigma_r^2(\bar{s}_k^i, a_k^i) + J_r(\bar{s}_k^i, a_k^i)\bar{\Sigma}_k^i J_r(\bar{s}_k^i, a_k^i)^\top + \gamma^2 \mathbb{V}_{\nu_{k+1}^i}[\nu_{k+1}^i]. \quad (11)$$

$\mathbb{V}_{\nu_{k+1}^i}[\nu_{k+1}^i]$ is computed iteratively backwards during the backup, starting from the last node to be expanded $k = T$, where equation 7 is used:

$$\mathbb{V}_{\nu_T^i}[\nu_T^i] \approx \sigma_v^2(\bar{s}_T^i) + J_v(\bar{s}_T^i)\,\bar{\Sigma}_T^i\,J_v(\bar{s}_T^i)^\top. \quad (12)$$

Approaches to incorporating OP2E into other search heuristics such as PUCT (which is used by MuZero) can be found in appendix B.1. Action selection in the environment can be done in the same manner as for exploitation (for example, sampling actions with respect to the visitation counts of each action at the root of the tree, in the original MuZero), but based on the exploratory tree.

### 3.3 Learning from Exploratory Decisions

If the policy executed in the environment is exploratory, in particular with exploration that is consistent over multiple time-steps, the targets generated by an algorithms with on-policy learning properties such as MuZero (see section 2.3) will reflect an exploratory policy and not the exploitation policy the agent is expected to learn. To alleviate this, we propose three main modifications to the training process which build off of each other: 1) separating training into alternating *exploration/exploitation episodes*, 2) *double planning* and 3) *max targets*. The extensions we propose enable generating value targets that are off-policy both online as well as during Reanalyses. In addition, we maintain some targets that are on-policy but yield better observed returns. These extensions are natural to combine with Reanalyse for further stabilizing of learning from exploratory trajectories.

**Alternating episodes**    First, we propose to alternate between two types of training episodes: *exploratory* episodes that follow an exploration policy throughout the episode, and *exploitatory* episodes that follow a reward-maximizing exploitation policy throughout the episode. This enables us to provide the agent with quality exploitation targets to evaluate and train the value and policy functions reliably, while also providing a large amount of exploratory samples, that explore the environment much more effectively and are more likely to encounter high-reward interactions earlier in the agent's training. In our experiments we used a ratio of one exploration episode per one exploitation episode, but other ratios as well as dynamically changing ratios are also natural to use. We leave an investigation into optimal exploration / exploitation ratios to future work.

**Double planning**    Second, we propose to compute two separate planning trees at each time step in exploration episodes: an exploration tree following the heuristic in Equation 10 and an exploitation tree following the regular UCT heuristic in Equation 1. When generating value targets from exploration trajectories, the value bootstrap $v_{t+n}^{\pi}$ used in the target is the value approximated by the tree planning with exploitation policy $\pi = \pi_\rho$ instead of tree planning with an exploration policy $\pi = \pi_\sigma$:

$$v_t^{target} = \sum_{i=0}^{n-1} \gamma^i r_{t+i}^{\pi_\sigma} + \gamma^n v_{t+n}^{\pi_\rho} \,. \tag{13}$$

This enables generating off-policy value-bootstraps $v_{t+n}^{\pi_\rho}$ online (similar to Reanlyse's process offline), but it's important to note that the rewards $\sum_{i=t}^{n-1} \gamma^i r_{t+i}^{\pi_\sigma}$ in the n-step value target are still the rewards observed in the environment following the exploration policy $\pi_\sigma$. While this doubles the computation cost of planning (or halves the computation budget for each planning tree), the results presented by Danihelka et al. (2021) showed that with a modified version of MuZero the contribution of a large planning budget reduces significantly, and with it, the expected consequence of halving it.

**Max targets**    Third, when generating value targets from exploration episodes, we propose to evaluate whether the exploratory interactions resulted in an improved return compared to the approximation of the MCTS (a *positive* exploration trajectory, left) or not (a *negative* exploration trajectory, right):

$$\sum_{i=t}^{n-1} \gamma^i r_{t+i}^{\pi_\sigma} + \gamma^n v_{t+n}^{\pi_\rho} > v_t^{\pi_\rho}, \qquad \sum_{i=t}^{n-1} \gamma^i r_{t+i}^{\pi_\sigma} + \gamma^n v_{t+n}^{\pi_\rho} \le v_t^{\pi_\rho} \,. \tag{14}$$

In positive exploration trajectories, the n-step exploratory value target $v_t^{n\text{-}step} = \sum_{i=t}^{n-1} \gamma^i r_{t+i}^{\pi_\sigma} + \gamma^n v_{t+n}^{\pi_\rho}$ can be used. In negative exploration trajectories, a zero-step exploitatory target can be used instead $v^{0\text{-}step} = v_t^{\pi_\rho}$. This is equivalent to generating the value targets with a max operator:

$$v_t^{target} = \max(v_t^{n\text{-}step}, v_t^{0\text{-}step}) \,. \tag{15}$$

Similarly, policy targets for exploratory episodes can be generated based on a max operator in a similar manner, where the policy used as the target is the policy that has achieved the largest value:

$$\pi^{target} = \arg\max_\pi (v_t^{n\text{-}step}, v_t^{0\text{-}step}) \,. \tag{16}$$

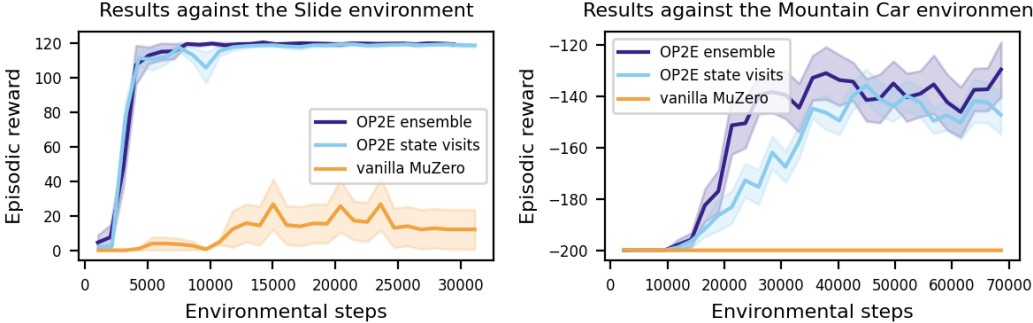

Figure 1: vanilla MuZero vs. OP2E with ensemble-variance uncertainty vs. OP2E with state-visitation-counting uncertainty against both environments. The performance of 10 independent seeds is averaged, and standard error of the mean (SEM) is presented in the shaded area. Both variants of OP2E are able to solve both tasks effectively while vanilla MuZero is not able to solve either of the tasks at all, on average, under the training-steps budget.

While this enables us to generate off-policy targets from negative exploration trajectories, the presence of 0-step value targets may cause other effects that are detrimental to learning. We explore some of the effects of this series of modifications in an ablation study in the experiments section. In addition, we note that the *max targets* approach only applies to environments with deterministic reward schemes.

## 4 EXPERIMENTS

To evaluate OP2E we compare the modified exploratory MuZero to vanilla MuZero against two hard-exploration tasks: Slide, a toy task similar to the Chain environment (Osband et al., 2016) to evaluate the extent to which deep exploration is achieved, as well as Mountain Car (Moore, 1990) which is a standard hard-exploration task used to evaluate advanced exploration approaches. In Slide, the environment consists of a chain of 60 states, with the only positive reward at the rightmost end of the chain and 0 rewards otherwise. We use a non-Markovian reward of $R(s_{59}, a_{right}) = \tau_{max} - \tau_{elapsed}$, the difference between the maximum time steps allowed by the environment and the number of elapsed time steps by the agent. The timeout is set to length of the environment times 3. The agent has access to three actions: left, right, stay. Left results in sliding back 10 positions, or otherwise to the starting position. Right results in proceeding to the right by 1 position. Stay results in sliding back 5 positions. This results in an environment that is very adversarial to exploration that relies on random action selection. In both tasks two variants of OP2E are evaluated: one with deep ensembles with prior functions (*OP2E ensemble*) for a scalable, realistic uncertainty estimation mechanism, and one with discretized state visitation counting (*OP2E state visits*) as an example of a reliable but unrealistic epistemic uncertainty estimation. As a proof of concept, and to avoid modifying MuZero more than absolutely necessary, we replace the state-uncertainty covariance function $\Sigma(s, a)$ with 0, i.e. ignored uncertainty of the forward model. With the exception of the exploration coefficient $C_\sigma$ (see appendix B.5) no dedicated tuning of hyperparamers was conducted for any of the variations or tasks used in the experiments. The results against both tasks are presented in Figure 1. OP2E is able to effectively solve both tasks, while vanilla MuZero is unable to solve the tasks at all under the training-steps budget provided.

### 4.1 ABLATIONS

An ablation study of the different modifications proposed for learning from exploratory trajectories in the Mountaincar environment are presented in Figure 2. The state-visitation counting uncertainty estimator (exploratory state visits) was used in all ablation experiments. In all ablations, the *max targets* agent is an agent with all modifications to training and target generation introduced in Section 3.3 applied: max value targets (eq. 15), max policy targets (eq. 16), alternating between exploration and exploitation episodes and double planning. In every ablation study only one modification is

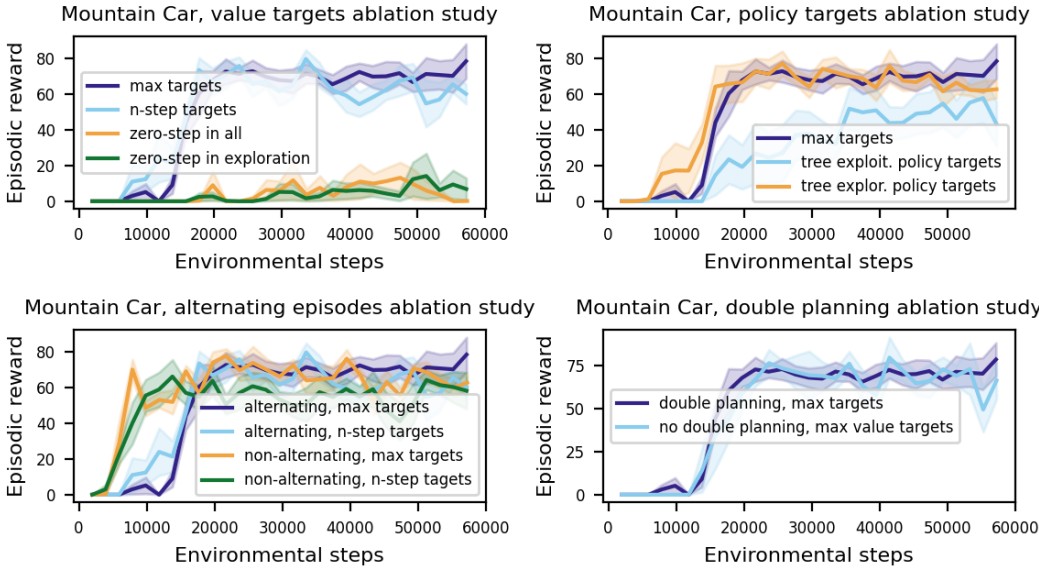

Figure 2: Target ablations against Mountain Car. The performance of 5 independent seeds is averaged, and standard error of the mean (SEM) is presented in the shaded area. A modified reward scheme is used for Mountain Car, see appendix B.4 for details.

investigated and all other modifications are left active. For example, in the *value targets ablation study* (top left of Figure 2) the value targets are varied, and in the *policy targets ablation study* (top right of Figure 2) the policy targets are varied, but all other design decisions remain the same as in Figure 1. *Value targets ablations* compare between four variants of value targets: max targets, n-step targets, 0-step targets in exploration episodes, and 0-step targets in all (both exploratory and exploitatory) episodes. *Policy target ablations* compare between three variants of policy targets: max targets, only-exploratory policy targets (policy targets in exploratory episodes are generated by the exploratory planning trees) and only-exploitatory policy targets (policy targets are generated from exploitatory planning trees in all episodes). *Alternating episodes ablations* compare between four variants: max value targets with alternating episodes, max value targets without alternating episodes (all episodes are exploration episodes), n-step value targets with alternating episodes, n-step value targets without alternating episodes (all episodes are exploration episodes). Finally, *double planning ablations* compares between an agent that uses double planning and an agent that does not (all episodes are exploratory, all value bootstraps and policy targets are exploratory).

In all experiments in the ablation study the *max targets* variant which uses all modifications proposed in section 3.3 dominates or matches the performance of all other variants in terms of final policy as well as learning stability.

While relying on 0-step value targets to turn MuZero into a completely off-policy algorithm is clearly not a conducive approach, Using max value targets demonstrates an ability to overcome this effect (Figure 2, top left). Generating policy targets based on exploitatory trees even in exploration episodes shows significant slowdown of learning (Figure 2, top right). This implies that using Reanalyse to update policy targets from exploratory trajectories with exploitatory policy targets, without utilizing the max operator to discern between positive and negative exploration trajectories, can slow down learning significantly. We expect such a slowdown to occur as a result of diverting the agent away from high return exploratory trajectories that differ significantly from the agent's current exploitation policy. Alternating between exploration and exploitation episodes showed significant slowing of learning to reach the goal state, while showing slightly better final policy and learning stability (Figure 2, bottom left). We expect the delay in learning to be a result of alternating episodes with a ratio of 1 to 1, and better ratios to be able to provide the same benefit with less drawback.

## 5 RELATED WORK

Plan2Explore (P2E, Sekar et al., 2020) is a previous approach that harnesses planning with learned models and uncertainty for advanced exploration, although in a different setting to the one tackled in this paper. P2E proposes pre-training of an exploration policy for a task, without access to a reward signal. The policy is trained to choose actions that maximize the epistemic uncertainty associated with the transition model. This policy is regularly updated with newly collected samples, until the entire state space is sufficiently explored and the learned model therefore precise enough everywhere to plan well. P2E only considers state prediction uncertainty and propagates the uncertainty through planning as intrinsic reward as opposed to variance. In addition, P2E requires an extended pre-training phase and does not scale to large state-spaces, because the learned exploration policy is trained to explore the entire state space without trading off for more promising areas with respect to expected return. In contrast, our approach propagates the uncertainty explicitly, enables a trade-off in real time between exploration and exploitation and is applicable to the standard online exploration setting in RL.

Epistemic uncertainty has developed into a standard for driving exploration in RL with methods such as Bootstrapped-DQN (Osband et al., 2016), the uncertainty Bellman-equation (UBE, O'Donoghue et al., 2018) and intrinsic reward (IR, Oudeyer & Kaplan, 2009). UBE approximates an analytically-motivated upper bound on the epistemic uncertainty associated with Q-values, in the form of a variance. The variance is used to drive exploration using Thompson sampling. UBE has not been developed explicitly for model-based approaches, and thus does not consider transition-based un-certainty, nor does it take advantage of the ability of online planning to propagate information from future decisions into current decisions immediately. In IR the epistemic uncertainty propagates di-rectly through the learned Q-value function, by learning a modified function:

$$Q_{IR}^\pi(s_t, a_t) = \mathbb{E}\big[r(s_t, a_t) + u(s_t, a_t) + \gamma V^\pi(s_{t+1})|s_{t+1} \sim P(\cdot|s_t, a_t)\big]$$

Where $u(s_t, a)$ is an uncertainty-bonus that is used to encourage the agent to explore less-visited transitions. Coupled with the limitations, a strength of propagating the uncertainty through value learning compared to through planning is that the horizon of uncertainty propagation is not lim-ited to the horizon of the planning (which is logarithmic in the planning budget of a planning tree in the worst case, and linear in the best case). Instead, the horizon is limited by the number of learning steps, which constrains the propagation differently. For example, deep Q-networks (DQN, Mnih et al., 2015) make one gradient update-step per environmental step, whereas planning always propagates immediately until the planning horizon. Moreover, these methods can be combined with our approach by using UBE to replace uncertainty-in-leaf value prediction in MuZero. Propagat-ing newly discovered uncertainty until the planning horizon happens immediately with our method, whereas longer delays are propagated through the value function.

## 6 CONCLUSIONS

By propagating epistemic uncertainty in real time through a planning tree together with planning for an exploratory objective, OP2E demonstrates deep exploration while circumventing issues arising from propagating uncertainty through value learning. In addition, the extensions we propose to learning from consistently-exploratory decisions show capacity to reach better final policies in ad-dition to a more stable learning process, effects which we expect to grow stronger in the presence of environments with more complex reward schemes and a more varied range of near-optimal poli-cies. The framework developed for approximating the epistemic uncertainty as it propagates in a planning tree can provide a backbone for additional approaches that take advantage of uncertainty estimation, such as planning for reliable execution (by discouraging uncertain decisions) or reliable planning (by avoiding planning into trajectories where additional planning *increases* uncertainty). The propagation framework itself is essentially modular from both MuZero and MCTS, making it an available choice for any approach that plans with a learned model and planning trees. Our ap-proach strengthens the notion that incorporating epistemic uncertainty into different online inference mechanisms can provide effective exploration by harnessing the strengths of the mechanism to make better exploratory decisions.

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
