# OpenReview forum: "Planning With Uncertainty: Deep Exploration in Model-Based Reinforcement Learning"
_ICLR.cc/2023/Conference — Submitted to ICLR 2023_

### Official Review · Reviewer_6utc · 2022-10-20

**Confidence:** 3
**Correctness:** 2
**Technical Novelty And Significance:** 3
**Empirical Novelty And Significance:** 2
**Recommendation:** 3

**Clarity, Quality, Novelty And Reproducibility:**

The paper is mostly clear, but there are some definitions that could be clarified. For example, the paper makes reference to a “somewhat on-policy” algorithm (Section 3.3), and it is not clear to me what that is. Usually, an RL algorithm is either on-policy, or off-policy; this is a binary classification.

Further, the paper defines deep exploration as “exploration that is able to propagate uncertainty from distant areas of the state-action space into local decisions and is consistent over multiple time-steps.” This explanation is confusing to me. Deep exploration could be more simply described as “exploration which is directed over multiple time steps” (Osband et al., 2016), or as exploration that takes “​​several coherent actions to explore unknown states instead of just locally choosing the most promising actions” (Parisi et al. 2022).

The content of the paper appears to be novel, and given the information provided in the paper I suspect that the results would be reproducible.

References:

Osband et al. (2016), Deep exploration via Bootstrapped DQN, NIPS 2016 (https://proceedings.neurips.cc/paper/2016/file/8d8818c8e140c64c743113f563cf750f-Paper.pdf)

Parisi et al. (2022).  "Long-Term Visitation Value for Deep Exploration in Sparse-Reward Reinforcement Learning" Algorithms 15, no. 3: 81. (https://doi.org/10.3390/a15030081)


**Strength And Weaknesses:**

The main strength of this paper is the extension to MCTS that propagates an estimate of uncertainty through the planning tree, and drives exploration in the environment. This idea is potentially very powerful if one has access to good measures of uncertainty.

The main weakness of this paper is the experimental evaluation of the claim that OP2E provides improved deep exploration over plain MuZero. The results provided in the two domains are not enough to convince me of this.

The Slide environment appears to be a genuinely hard exploration task, even though it is a tabular MDP. Results in this domain are good, and support the conclusions. The environment is described as giving a positive reward when the agent reaches the rightmost state. What is the magnitude of this reward? (This would help with my understanding of Figure 1.)

I strongly disagree that Mountain Car is a “hard-exploration task used to evaluate advanced exploration approaches.” I think evaluating an exploration algorithm on Mountain Car does not provide much insight. Sarsa with a simple linear function approximation scheme and epsilon-greedy exploration can learn a good policy in less than 500 episodes (p. 246, Sutton & Barto, 2018). This is significantly less than ~20,000 episodes required to learn a good policy for OP2E. Mountain Car is at best a validation test to ensure that the components of an algorithm are working correctly, but not as evidence of advanced deep exploration.

I suggest that the authors test their algorithm on a few more hard exploration problems. Ideally on mix of problems that are tabular and that require function approximation. Parisi et al. (2022) describe many such problems, like Deep Sea for example.


References:

Sutton and Barto. (2018). Reinforcement learning: an Introduction. 2nd Ed. MIT press.

Parisi et al. (2022).  "Long-Term Visitation Value for Deep Exploration in Sparse-Reward Reinforcement Learning" Algorithms 15, no. 3: 81. (https://doi.org/10.3390/a15030081)


**Summary Of The Paper:**

This paper presents a new extension of the MuZero algorithm (called online planning to explore, or OP2E) that leverages estimates of uncertainty in the algorithm’s predictions of value and reward to generate exploratory behavior. This approach has the advantage of propagating the uncertainty estimates to the policy through the planning tree, rather than through a value function like in other methods. The paper claims that OP2E is better able to achieve deep exploration, than standard MuZero. The claims are supported by experimental results in two different domains.


**Summary Of The Review:**

Even though the proposed algorithm appears to be novel and interesting, there are some flaws in the experimental evaluation. For this reason I suggest rejecting this paper.

The paper could be improved by expanding the experimental evaluation to include some additional environments that require deep exploration, and/or a larger more complex environment that requires deep exploration.

---

> ### Author Response · Authors · 2022-11-12
> **Reply to reviewer 6utc**
>
> Dear reviewer,
>
> Thank you for the review.
>
> We agree that these two environments are not sufficient evidence that OP2E will be sufficient to explore harder settings (such as Atari's Montezuma's Revenge).
> Unfortunately, we do not believe we will be able to provide additional results on more complex environments such as Atari within this rebuttal, but we would like to stress the large improvement our method shows even on toy examples.
>
> In regards to the comments and questions:
>
> Slide provides a non-Markovian reward at the goal: $ r_{goal} = \tau_{max} - \tau_{elapsed} $. $ \tau_{max} $ is number of allowed steps before timeout and $ \tau_{elapsed} $ is number of elapsed steps.
> $ \tau_{max} $ is set to x3 the length of the environment, and induces a reward of x2 the length of the environment for the optimal policy (120 with the environment length of 60 used in our experiments).
>
> Mountain Car is generally considered a hard exploration problem. The cited result from Sutton and Barto (2018), p.246, extended the episode length to 1000 and used optimistic initialization, a technique that works for tile-coding, but not for neural networks (Rashid et al., 2020). We test our method with the default 200 time steps, which is very challenging, as the optimal trajectory takes around 110 time steps and all approaches with random (epsilon greedy) exploration fail here.
>
> In regards to on / off-policy-ness:
> As discussed in the MuZero-Reanalyse paper (Schrittwieser et al., 2021, the last paragraph of section 4), MuZero's learning of policy and reward can be considered off-policy, but MuZero's learning of value that uses n-step targets can be considered as on-policy, resulting in learning that is for the most part off-policy, but not entirely.
> We discuss this in more detail in our paper at the end of section 2.3.
>
> REFERENCES
>
> Rashid, T., Peng, B., Boehmer, W., and Whiteson, S. "Optimistic Exploration with Pessimistic Initialisation." Proceedings of the International Conference on Learning Representations. 2020.
>
> J. Schrittwieser, T. Hubert, A. Mandhane, M. Barekatain, I. Antonoglou,
> and D. Silver, “Online and offline reinforcement learning by planning with a
> learned model,” Advances in Neural Information Processing Systems, vol. 34,
> pp. 27 580–27 591, 2021.

---

> > ### Comment · Reviewer_6utc · 2022-11-16
> > **Reply to rebuttal**
> >
> > Dear reviewers,
> >
> > Thank you for answering my questions and providing clarifications.
> >
> > However, because of the experimental evaluation, I have to maintain my current recommendation.

---

### Official Review · Reviewer_3Uzt · 2022-10-23

**Confidence:** 4
**Correctness:** 4
**Technical Novelty And Significance:** 3
**Empirical Novelty And Significance:** 3
**Recommendation:** 3

**Clarity, Quality, Novelty And Reproducibility:**

This paper is well-written and easy to follow. Adding a UCB type of exploration bonus is not a novel idea, but it is good to have some practical implementation of the theoretical results. The reviewer also believes the experiments are reproducible.

**Strength And Weaknesses:**

Strength:
1. This paper is well-written and easy to follow.
2. This proposed method adopts the classic UCB exploration bonus from RL theory, which is intuitive and the reviewer appreciates the efforts towards connecting RL theory and practice.
3. The experimental results are strong, among all tested environments.

Weakness:
1. As an empirical paper, the reviewer thinks the experimental results are not enough to demonstrate the empirical benefits of the proposed method – the two tested environments are too toy which limits the contribution of this work.


**Summary Of The Paper:**

This paper proposes a modified version of MCTS that encourages exploration and incorporates uncertainty into the planning tree. The authors compare the performance of the proposed method against MuZero in two environments: Slide and Mountain Car.

**Summary Of The Review:**

The reviewer highly appreciates this work as it is trying to implement the classic UCB type of exploration bonus to the powerful MuZero framework. But at the current moment, the reviewer is having a hard time accepting this paper based on empirical evidence of two toy environments. The reviewer understands that some of the environments in the MuZero paper require large computational resources, however, perhaps the author can still try the proposed method in some easy games in Atari that require a reasonable amount of computation to make the empirical results stronger and more convincing. The reviewer would like to change the rating once the author can provide more empirical evidence of the proposed method in more challenging tasks (say Montezuma in Atari).

---

> ### Author Response · Authors · 2022-11-12
> **Reply to reviewer 3Uzt**
>
> Dear reviewer,
>
> Thank you for the review.
>
> We agree that these two environments are not sufficient evidence that OP2E will be sufficient to explore harder settings (such as Atari’s Montezuma’s Revenge). Unfortunately, we do not believe we will be able to provide additional results on more complex environments such as Atari within this rebuttal, but we would like to stress the large improvement our method shows even on toy examples.

---

> > ### Comment · Reviewer_3Uzt · 2022-11-13
> > **Reply to rebuttal**
> >
> > Dear Authors,
> >
> > Thank you for the replies. Based on the current experiment, I have to maintain my recommendation on this work. I wish all the best for this work and I am looking forward to seeing how the proposed methods scale up to larger environments.

---

### Official Review · Reviewer_YegC · 2022-10-24

**Confidence:** 4
**Correctness:** 2
**Technical Novelty And Significance:** 3
**Empirical Novelty And Significance:** 2
**Recommendation:** 3

**Clarity, Quality, Novelty And Reproducibility:**

The work represents an interesting extension to tree search methods within MBRL, which directly addresses exploration through uncertainty bonuses. While not entirely novel, it represents an important are to research. The work is clear, but its quality is held back by an insufficient experimental evaluation.

**Strength And Weaknesses:**

Strengths:

- The paper is well written, the motivation is clearly states and contributions are thoroughly explained.
- While the training procedure introduces several modifications (alternating between exploration and exploitation, using exploitation value bootstraps and max targets), an appropriate ablation study is presented in Figure 2.

Weaknesses:
- The experimental evaluation of OP2E is limited to a selection of problems which is small (2 environments), and only includes toy problems. On the other hand, OP2E is presented as generally applicable in problems for which MuZero was designed. As a result, an evaluation on a larger number of challenging environments (e.g. standard exploration-hard games in ATARI), would be extremely helpful in improving this paper's soundness.
- The paper argues that naively propagating uncertainty through the value function suffers from several problem (p. 1, last paragraph). While such arguments are well supported, they should also be validated empirically if possible.

Minor comments:

- Section 2.2: the objective of MCTS is conventionally to find $a = \argmax Q^\pi(s_t, a)$ rather than a policy maximizing each action-state value.
- Section 2 is very detailed, and can be shortened significantly to create space for experimental results.
- Eq. 4: is $\mathbb{E}_{s_k} [\mathbb{E}_{s_{k+1}}[s_k | s_k, a_k]]$ supposed to be $\mathbb{E}_{s_k} [\mathbb{E}_{s_{k+1}}[s_{k+1} | s_k, a_k]]$ instead?
- Section 4: restricting the experimental evaluation to a deterministic world model significantly simplifies the uncertainty propagation mechanism. Although this is acknowledged, empirical results could benefit from an evaluation of stochastic forward models.
- Section 5 could include relevant work on MuZero in stochastic settings [1].

References:
[1] Antonoglou et al. "Planning in Stochastic Environments with a Learned Model" in ICLR 2022

**Summary Of The Paper:**

This paper identifies lackluster exploration as a standing problem in model-based deep RL algorithm involving tree search. Hence, the authors propose to estimate epistemic uncertainty in order to drive exploration.  Instead of directly integrating uncertainty bonuses in the value function, their method directly propagates uncertainty through the planning tree, thus decoupling uncertainty and value estimation. The paper brings forth three main contributions: (1) a propagation scheme for uncertainty through a planning tree, (2) a modification of the planning objective that includes epistemic uncertainty and encourages exploration and (3) a decoupled training process for exploratory and exploitative episodes, which is designed to stabilize training. The technique described in (1) enables efficient computation of propagated entropy by using Taylor approximations and independence assumptions between reward  at time step $k$ and n-step returns at time step $k+1$. (2) consists in adding an uncertainty bonus to the action selection criteria in the tree policy in order to encourage global exploration. Finally, the training process from (3) relies on alternating exploration and exploitation episodes. The former compute two planning trees in parallel: actions are selected from a tree with uncertainty bonus, while value bootstrap terms are obtained by a search procedure not involving uncertainty bonuses. Finally, policy and value utilize n-step exploratory targets only when greater than exploitation value estimates. The authors provide a brief experimental evaluation against vanilla MuZero, and an ablation study on toy environments (Slide, Mountain Car).

**Summary Of The Review:**

The paper tackles an important problem by combining two interesting ideas (MBRL with tree search, and uncertainty-driven exploration) in a principled way. The quality of the paper is good, and the main contributions are clearly presented, but they are only validated on a limited number of toy problems. The lack of convincing experiments in challenging domains constitutes the main weakness of this paper, and I recommend rejection in its current state.

---

> ### Author Response · Authors · 2022-11-12
> **Reply to reviewer YegC**
>
> Dear reviewer,
>
> Thank you for the review.
>
> We agree that these two environments are not sufficient evidence that OP2E will be sufficient to explore harder settings (such as Atari’s Montezuma’s Revenge). Unfortunately, we do not believe we will be able to provide additional results on more complex environments such as Atari within this rebuttal, but we would like to stress the large improvement our method shows even on toy examples.
>
> Thank you for the additional comments, we will incorporate them into the paper.

---

> > ### Comment · Reviewer_YegC · 2022-11-17
> > **reply to rebuttal**
> >
> > Dear Authors,
> >
> > thank you for your response. I will keep my rating and hope you can provide more evidence for a future submission.

---

### Official Review · Reviewer_u2v3 · 2022-10-25

**Confidence:** 3
**Correctness:** 3
**Technical Novelty And Significance:** 2
**Empirical Novelty And Significance:** 2
**Recommendation:** 5

**Clarity, Quality, Novelty And Reproducibility:**

**Clarity**
The presentation is clear.

**Quality**
The quality of the paper is good.

**Novelty**
The proposed method is novel.

**Reproducibility**
Need more details about the algorithms and experiments to reproduce the results.


**Strength And Weaknesses:**

**Strength**

+ This paper modifies Monte Carlo Tree Search (MCTS) for an exploratory objective, where the best actions are predicted for gathering relevant information rather than for highest expected return. This was done by first propagating uncertainty in a plannig tree, and then optimizing the action selection for exploration using a modified UCT operator with the propagated uncertainty.

+ Propagating uncertainty in planning tree is novel and interesting.

**Weakness**

+ Need to compare with more exploration baselines.

+ Need to evaluate the proposed method on more diverse and challenging tasks.

Questions:

+ Can the proposed method extend to other model-based RL methods?


**Summary Of The Paper:**

This paper proposes planning with uncertainty, which incorporates epistemic uncertainty into planning trees for deep exploration in model-based RL. Built on top of the SOTA model-based RL algorithm MuZero, the proposed method has shown effectiveness of exploration with standard uncertainty estimation methods.

**Summary Of The Review:**

Overall, the proposed method is novel and interesting, but need to further strengthen the empirical evaluation.

---

> ### Author Response · Authors · 2022-11-12
> **Reply to reviewer u2v3**
>
> Dear reviewer,
>
> Thank you for the review.
>
> We aim to illustrate that incorporating uncertainty into planning is sufficient to enable deep exploration, and because of this we opted not to compare with other baselines.
>
> We agree that these two environments are not sufficient evidence that OP2E will be sufficient to explore harder settings (such as Atari’s Montezuma’s Revenge).
> Unfortunately, we do not believe we will be able to provide additional results on more complex environments such as Atari within this
>  rebuttal, but we would like to stress the large improvement our method shows even on toy examples.
>
> In regards to the question: we believe that OP2E is extensible to methods that use a model and planning trees. Methods that pass gradients through the model might also be applicable using second-order gradients on the Jacobians. We discuss this briefly in the last two sentences of the conclusion.

---

### Decision · Program_Chairs · 2023-01-20

**Decision:**

Reject

**Justification For Why Not Higher Score:**

All the reviewers agreed this paper tackles an important problem and that the proposed method was sensible and compelling. However, the reviewers also found the evaluation lacking, as OP2E was only evaluated on two relatively small domains. While I find the results on these domains to be impressive, I agree with the reviewers that the approach should be evaluated in a wider variety of domains. I agree with the authors in their response that the domains they've tested are challenging from an exploration standpoint, but they fail to demonstrate the generality of OP2E as they are both relatively simple and tabular (even if challenging) and small in number. It is therefore difficult to evaluate whether OP2E would work well in general on the types of tasks that MuZero has previously been evaluated on (i.e., whether it's something that should generally just be incorporated into MuZero or not).

I think this paper would be a great workshop paper, but without the additional experiments I do not think it quite reaches the bar for publication at ICLR.

**Justification For Why Not Lower Score:**

N/A

**Metareview: Summary, Strengths And Weaknesses:**

This paper proposes a modification to the MuZero algorithm to support deep exploration (referred to as OP2E) by incorporating epistemic uncertainty into planning trees. The paper presents results on two domains, Slide and Mountain Car, demonstrating that their exploratory version of MuZero strongly outperforms vanilla MuZero. The paper also presents an extensive ablation study examining the various choices of the algorithm.

All the reviewers agreed this paper tackles an important problem and that the proposed method was sensible and compelling. However, the reviewers also found the evaluation lacking, as OP2E was only evaluated on two relatively small domains. While I find the results on these domains to be impressive, I agree with the reviewers that the approach should be evaluated in a wider variety of domains. I agree with the authors in their response that the domains they've tested are challenging from an exploration standpoint, but they fail to demonstrate the generality of OP2E as they are both relatively simple and tabular (even if challenging) and small in number. It is therefore difficult to evaluate whether OP2E would work well in general on the types of tasks that MuZero has previously been evaluated on (i.e., whether it's something that should generally just be incorporated into MuZero or not). If the authors can include experiments on a wider range of domains (especially non-tabular ones that are considered canonically hard exploration problems, e.g. Montezuma) demonstrating the efficacy of the approach, then this would make a great contribution to the machine learning community. I hope that the authors will be able to include such experiments in a future revision of the paper.

**Summary Of Ac-Reviewer Meeting:**

N/A